# An Enhanced Indoor Positioning Algorithm Based on Fingerprint Using Fine-Grained CSI and RSSI Measurements of IEEE 802.11n WLAN

**DOI:** 10.3390/s21082769

**Published:** 2021-04-14

**Authors:** Jingjing Wang, Joongoo Park

**Affiliations:** School of Electronic and Electrical Engineering, Kyungpook National University, 80 Daehak-ro, Buk-gu, Daegu 41566, Korea; wjj0219@naver.com

**Keywords:** indoor fingerprinting localization, Gaussian filter, Kalman filter, received signal strength indicator, channel state information

## Abstract

Received signal strength indication (RSSI) obtained by Medium Access Control (MAC) layer is widely used in range-based and fingerprint location systems due to its low cost and low complexity. However, RSS is affected by noise signals and multi-path, and its positioning performance is not stable. In recent years, many commercial WiFi devices support the acquisition of physical layer channel state information (CSI). CSI is an index that can characterize the signal characteristics with more fine granularity than RSS. Compared with RSS, CSI can avoid the effects of multi-path and noise by analyzing the characteristics of multi-channel sub-carriers. To improve the indoor location accuracy and algorithm efficiency, this paper proposes a hybrid fingerprint location technology based on RSS and CSI. In the off-line phase, to overcome the problems of low positioning accuracy and fingerprint drift caused by signal instability, a methodology based on the Kalman filter and a Gaussian function is proposed to preprocess the RSSI value and CSI amplitude value, and the improved CSI phase is incorporated after the linear transformation. The mutation and noisy data are then effectively eliminated, and the accurate and smoother outputs of the RSSI and CSI values can be achieved. Then, the accurate hybrid fingerprint database is established after dimensionality reduction of the obtained high-dimensional data values. The weighted k-nearest neighbor (WKNN) algorithm is applied to reduce the complexity of the algorithm during the online positioning stage, and the accurate indoor positioning algorithm is accomplished. Experimental results show that the proposed algorithm exhibits good performance on anti-noise ability, fusion positioning accuracy, and real-time filtering. Compared with CSI-MIMO, FIFS, and RSSI-based methods, the proposed fusion correction method has higher positioning accuracy and smaller positioning error.

## 1. Introduction

With the development of wireless communication technology, the demand for location-based services (LBS) has increased greatly. The complex indoor environment makes GPS [1,2,3,4] signals that belong to the outdoor positioning system vulnerable to multi-path effects, which makes it impossible to apply in indoor positioning. The GPS [1,2,3,4] signal is blocked or reflected by the wall, so the satellite signal cannot be received in the indoor environment. Therefore, GPS cannot be used for positioning in an indoor environment. Common indoor positioning technologies include ultra-wideband (UWB), infrared, radio frequency identification (RFID), ultrasonic, LED visible light, ZigBee, Bluetooth, WiFi, geomagnetic, etc. There are many localization methods for wireless sensor networks. At present, these methods can be categorized into the followings: distance/angle-based positioning algorithm (Range-Based) and distance-independent positioning algorithm (Range-Free). The representative range-based positioning algorithms are time of arrival (TOA) [5], time difference of arrival (TDOA) [6], angle of arrival (AOA) [7], and received signal strength indication (RSS) [8,9]. On the other hand, representatives of range-Free positioning algorithms are centroid algorithm, DV-Hop algorithm, Amorphous algorithm, APIT algorithm, etc. The performance evaluation criteria for a positioning algorithm mainly include positioning accuracy, node density, fault tolerance and adaptability, power consumption, cost, etc. Among them, the widespread deployment of indoor WiFi wireless access point (AP) has rendered the WiFi positioning technology feasible, and the location method based on location fingerprint is becoming the mainstream of indoor positioning due to its low cost and simple method. The traditional fingerprint location method usually includes two phases: offline training phase (offline phase) and online prediction phase (online phase). In the offline phase, the wireless signal characteristics of each reference point Reference Point (RP) are collected as fingerprints, and a database is established. In the online stage, by matching the wireless signal fingerprint characteristics of the user’s location with the database information, the location corresponding to the fingerprint with the highest matching degree is obtained as the user’s final location. Most of the existing fingerprint location systems use RSSI as coarse-grained fingerprint information, which is simple and does not need additional equipment configuration.

IEEE 802.11 protocol does not give specific processes or algorithms to generate RSSI. RSS refers to the MAC layer signal strength received by the client. Because RSS is coarse-grained information, it is often affected by multi-path effect and noise signal, and the location performance is not stable, so it can not meet the requirements of indoor precise location. In recent years, CSI [10,11] can be obtained through commercial Wi Fi devices. CSI is the channel state information that measures the channel condition. It belongs to PHY layer and comes from the sub-carriers decoded in Orthogonal Frequency Division Multiplexing (OFDM) [12] system. CSI is fine-grained physical information, which is more sensitive to the environment, so it is applied in the fields of action recognition, gesture recognition, tracking, and so on. The CSI based on the physical layer makes up for the shortcomings of the average superimposed amplitude processing method in the traditional RSSI. The phase information of each sub-carrier is added to provide more precise and stable signal characteristic information for indoor WiFi fingerprint positioning technology. CSI can minimize the effects of multi-path and noise by analyzing the transmission of different sub-channel signals. CSI has opened up a new space for the WiFi-based indoor positioning technology and the research effort has been directed to this field.

In Reference [13], CSI amplitude information is used for positioning and important fingerprint features. The phase information is not used. A positioning system Pinloc is proposed in Reference [14]. Pinloc estimates the distribution of a single sub-carrier in the complex plane through the probability density function, and then realizes the position estimation. However, the Pinloc system only uses the frequency diversity of CSI and does not consider the effect of spatial diversity. Reference [15] proposed a data processing method to reduce the phase deviation, but this paper did not consider the combined effect of random phase deviation and random delay deviation. Reference [16] also designed a fine-grained indoor fingerprint location system FIFS using CSI. Although FIFS utilizes both frequency diversity and spatial diversity, this method only aggregates the power of all sub-channels. The CSI-MIMO positioning system [17] only considers the amplitude and phase information of each sub-carrier. However, CSI-MIMO only considers spatial diversity and frequency diversity when generating location fingerprints, neglecting the characteristics of multiple receiving antennas, and the uniqueness of the location can not be reflected further. In recent years, the industry has proposed many fingerprint-based positioning methods, using different features (such as RSSI and CSI) as fingerprints to distinguish different locations. Although CSI-based methods generally have higher accuracy than RSSI-based methods, it is found that the positioning results of different methods usually compensate each other. By fusing different features, a more accurate positioning result can be obtained than a single feature. Following these backgrounds, this paper proposes a high-precision fingerprint positioning method based on RSSI and CSI.

Currently, most methods based on CSI fingerprints do not incorporate RSS information, thereby reducing the computing resources required for positioning. Therefore, this paper uses RSS and CSI, which are two kinds of information with different granularity, to realize regional location and precise location, respectively, and to make full use of the advantages of different granularity information. Compared with the distance estimation method, the complex indoor environment has less influence on fingerprint method. Based on the above reasons, this paper studies the indoor location method based on the hybrid fingerprint of RSS and CSI. Combining the respective characteristics of RSS information and CSI information, this paper proposes an improved fingerprint precise positioning algorithm based on RSSI and CSI. In the offline phase, the mobile terminal collects the RSSI signal and CSI signal from each AP in the known reference position. After denoising the original RSSI signal and CSI amplitude value by using the Kalman filter, the phase of CSI is preprocessed to build a more robust position fingerprint database. At the same time, to reduce the complexity of the fingerprint data and positioning error, we use the weighted k-nearest neighbor (WKNN) algorithm [18] to perform fingerprint matching and find the position coordinates in the online stage. Analyzing the experimental results shows that the improved fingerprint positioning algorithm has higher positioning accuracy in indoor environments.

The main contributions of this paper can be divided into the following:This paper proposes a novel cross-layer approach including MAC layer information and physical layer information that enables fine-grained indoor fingerprint location algorithm in OFDM-MIMO WLANs.The obtained RSSI value and CSI amplitude value are denoised, and CSI phase value is linearly transformed. The processed measurements information can express the difference of fingerprints between different locations.The proposed algorithm reduces the dimension of the amplitude and phase values of CSI, and constructs a fingerprint database that can map the location feature data.In this paper, an indoor fingerprint location method based on RSSI and CSI in high load AP environment is proposed. It improves the difficulty of getting RSS and CSI information of AP in high load WiFi channel due to beacon delay. The proposed method can be used in a high-load AP environment.The positioning accuracy of the proposed method in two typical indoor environments is high. This method is higher than several traditional localization algorithms, and it is a more accurate WLAN Indoor fingerprint location algorithm.

This article includes the following chapters: Section 2 introduces RSSI, CSI, and WKNN. In Section 3, the proposed fingerprint location method is described. We process the original RSSI and CSI information, respectively, and obtain the effective fingerprint database in the offline stage. We use the WKNN algorithm to estimate the location through the fingerprint database established by the effective RSSI and CSI. Section 4 analyzes the experimental results and discusses the performance of the algorithm. Finally, Section 5 is conclusions and future work.

## 2. Related Work

### 2.1. Characteristics of RSSI

RSSI refers to the received signal strength received by the client, which belongs to the MAC layer and comes from each packet. The multi-path propagation of wireless signal refers to the reflection, diffraction, and scattering of electromagnetic waves on the propagation path. The signal received by the node does not come from a single path but from multiple paths. Due to the different path distances of electromagnetic waves, the arrival time of nodes is different, and there is phase difference. The positive and negative superposition of different phase differences will enhance or reduce the original signal, resulting in multi-path attenuation, which makes the indoor electromagnetic environment present regional and special. Therefore, the channel multi-path structure is unique for each location, which is called the RSSI position “fingerprint” [19,20].

WLAN vendors can privately define RSSI values. Through the description in the 802.11-2007 standard [21], we know that RSSI is measured by the receiving network card, and there is path attenuation in the middle, so the sender can not determine the specific receiving power of the receiver. The process or algorithm of RSSI generation is not given in the 802.11 protocol. According to the protocol, the RSSI value ranges from 0 to 255, and the RSSI value increases monotonically with the energy of PHY preamble. Therefore, only the PHY preamble measurement can be selected as the RSSI value. The instantaneous value of RSSI is obtained by integrating baseband IQ power. The theoretical calculation formula of RSSI is as follows:(1)$$RSSI_{Instan}=\sqrt{I^{2}+Q^{2}}$$

The average value of RSSI within 1 s is obtained by averaging the instantaneous values of 8192 RSSIs. The average value of RSSI is calculated as follows:(2)$$RSSI_{Ave}=\frac{\sum_{i=1}^{8192}RSSI_{Instan}}{8192}$$

Power is sampled directly in time domain. Since most wireless signals are MW level, they are polarized and converted into dBm, which does not mean that the signal is negative.

### 2.2. Channel State Information Amplitude and Phase

CSI data belongs to the physical layer information of wireless communication protocol. The physical layer (PHY) of 802.11 protocol [17] is the interface between MAC and wireless media. CSI can be represented by the value of each element in the channel gain matrix H [22,23]. Channel impulse response (CIR) can describe the multi-path effect of wireless channel. It can be expressed in the following formula.
(3)$$H_{k}=∥H_{k}∥e^{j∠H_{k}}$$
where $H_{k}$ is the CSI of the kth sub-carrier. $∥H_{k}∥$ and $∠H_{k}$ represent amplitude and phase of kth sub-carriers, respectively. *H* appears in a complex form a+bi. CSI represents the coefficient of a wireless channel. We can get the modulus $\sqrt{\left(a^{2}+b^{2}\right)}$ and argument $\theta =argtan\frac{b}{a}$ of the complex number, that is, the corresponding amplitude and phase.

Channel frequency response (CFR) [24] can be used to describe transmission by amplitude frequency and phase frequency. And because CFR and CIR are Fourier transforms for each other in the case of infinite bandwidth, CFR can be expressed as follows.
(4)$$h\left(\tau \right)=\sum_{i=1}^{N}a_{i}e^{-j\theta_{i}}\delta \left(\tau -\tau_{i}\right)$$
where $a_{i}$ is the amplitude attenuation of the ith path. $\theta_{i}$ is the phase offset of the ith path. $\tau_{i}$ is the time delay of the ith path. *N* is the total number of paths propagated. δ(τ) is a Dirac impulse function.

Since CSI is the frequency response of multiple sub-carriers, it can accurately describe frequency selective channels. According to the Fourier transform of CIR and CFR, multiple propagation paths can be distinguished in time domain.

### 2.3. Comparison of CSI and RSSI

RSSI is the superposition of multiple path signals, which is very unstable. In a certain range, the probability of RSSI coincidence in different locations is relatively large, which makes it difficult for RSSI to complete high-precision indoor positioning, and it is only suitable for rough estimation of positioning range. CSI is not the superposition of all the sub-carrier information, it describes the signal of multiple paths, has more characteristics, and contains the channel state information of multiple sub-carriers. The amplitude and phase of CSI sub-carriers in different positions are crossed, so single sub-carrier cannot be distinguished. But if we use the diversity of CSI frequency and all the sub-carrier data, through a certain algorithm, we can distinguish different locations. Table 1 shows the differences between RSSI and CSI.

### 2.4. Weighted K-Nearest Neighbor (WKNN) Algorithm

The K-nearest neighbor (KNN) is a deterministic algorithm. It obtains the fingerprint of the first *K* reference points which are closest to the location point in space and calculates the location estimation value of the *K* reference points according to the coordinates of the *K* reference points. The basic distance space can be calculated as Equation (Equation 5).
(5)$$d_{q}={\left(\sum_{i=1}^{n}{\left|s_{i}-S_{i}\right|}^{q}\right)}^{\frac{1}{q}},$$
where $s_{i}$ is the RSSI from the positioning phase. $S_{i}$ is the RSSI from the fingerprint database. The variable *q* depends on the distance preferred by the algorithm. When q=2, it represents Euclidean distance. Although it is not accurate compared with the probabilistic algorithm, the KNN is one of the most popular algorithms because of its low computational complexity.
(6)$$\left(\hat{x},\hat{y}\right)=\frac{1}{k}\sum_{i=1}^{k}p_{i}\;p_{i}\in D_{1:k}.$$

The weighted k-nearest neighbor (WKNN) [25] algorithm was proposed to improve the KNN algorithm. KNN algorithm is to calculate the mean value of the coordinates of K reference points, while WKNN multiplies each reference point by a weighted coefficient, and then G weights and averages K reference points to obtain the location coordinates of the point to be located, as shown in Equation (Equation 7).
(7)$$\left(\hat{x},\hat{y}\right)=\sum_{i=1}^{k}\frac{\frac{1}{d_{i}+\varepsilon }}{\sum_{j=1}^{k}\frac{1}{d_{j}+\varepsilon }}\cdot \left(x_{i},y_{i}\right),$$
where $d_{i}$ represents the spatial distance between the fingerprint data of the ith reference point and the data measured in the online stage, ε is a positive number tending to 0, and the denominator in the formula is 0. $\left(x_{i},y_{i}\right)$ is the position coordinate of the ith reference point, and $\left(\hat{x},\hat{y}\right)$ is the position coordinate of the point to be located.

## 3. Proposed Indoor Fingerprint Localization Architecture and Methodology

To ensure location accuracy, this paper proposes an indoor hybrid fingerprint location algorithm based on RSSI and CSI. Because of the influence of multi-path effect, AP loss, heterogeneous network interference, and other factors, the collected fingerprint data can not be directly used, and this paper will carry out the relevant data processing work. This part first introduces the structure of the indoor fingerprint positioning model. Secondly, it describes the process of RSSI and CSI data fusion after the establishment of the fingerprint database.

### 3.1. Indoor Fingerprint Localization Architecture

The indoor fingerprint location method proposed in this paper includes two parts: offline phase and online phase. In the offline phase, a modified device supporting OFDM-MIMO is used to collect RSSI and CSI measurement data at the reference point. Then, according to the proposed method, the abnormal packets which significantly deviate from the whole in the continuous RSSI and CSI packets are removed. According to the location fingerprint generation method designed in this paper, the position fingerprint containing RSSI, CSI amplitude, and CSI phase information is generated. The fingerprint uses the characteristics of frequency diversity and space diversity of the Orthogonal Frequency Division Multiplexing (MIMO) system to obtain CSI and RSSI values on different antennas. In each sub-carrier, RSSI and CSI are aggregated in the unit of receiving antenna. At the same time, the anti-jamming ability and stability of the location fingerprint are further enhanced and the uniqueness of the location is better reflected, which helps to improve the accuracy of the location. Next, the location fingerprints generated in each reference location are collected to construct the location fingerprint map and database. In the offline phase, the coordinates of the target location are obtained by matching the fingerprint database. Figure 1 shows the flow chart of the positioning system structure.

### 3.2. Proposed Indoor Fingerprint Localization Methodology

This section describes the fingerprint generation process of the fusion location method proposed in this paper. The features of the RSSI are its low characteristic dimension and easy obtainability, however, it also suffers from poor signal stability and low spatial resolution. On the other hand, the CSI retains the characteristics of fine signal granularity, high stability, high characteristic dimension, and high computational complexity. Based on the above characteristics, a fingerprint fusion localization method based on RSSI and CSI is proposed. Table 2 shows the available information about CSI [26]. It can be seen from Table 1 that the CSI obtained contains RSSI values on different antennas. To obtain more accurate positioning, we use MATLAB to obtain the RSSI value on the receiver antenna. According to the mathematical expressions of RSSI and CSI in time domain and frequency domain and their corresponding characteristics, RSSI and CSI are compared and preprocessed.

#### 3.2.1. Processing of Raw RSSI Based on Gaussian-Kalman Filter

For indoor environment, the multi-path effect is obvious and RSS is more unstable. There may be some abnormal values in the measurement results of RSSI packets. If these abnormal values are used as the received signal strength, the positioning results will greatly deviate. The indoor location algorithm based on fingerprint similarity comparison needs to take the whole RSS vector as a whole. The fluctuation and noise of an RSS signal will have a great impact on the overall comparison results. We first process the RSSI value using Gaussian filtering [27]. The basic principle of Gaussian filtering is to establish a Gaussian distribution model for numerical values. Gaussian function is introduced in this method, and Gaussian function is discretized. The Gaussian function value at the discrete point is the weight value, that is, the RSSI values at the high probability occurrence regions are selected. The Gaussian filter can effectively filter the data significantly deviated from the true values and suppress the positioning error caused by signal mutation, but it is not effective in dealing with shadow effect, energy reflection, and other interference problems. Next, to smooth the output of RSSI data, this paper also uses the Kalman filter algorithm [28,29]. Kalman filtering can effectively filter indoor interference noise that obeys a normal distribution. After the Kalman filtering, the smoothed RSSI value of the reserved part is taken as the mean value and the final effective RSSI value is obtained. The above analysis shows that the proposed Gaussian-Kalman linear filter can obtain a more effective measurement value RSSI.

In this paper, the Dell notebook modified the wireless network card was used for data collection at Kyungpook National University (KNU), and the RSSI on different antennas was obtained. The detailed process of the experimental environment and data collection is described in detail in Section 4.1. Figure 2 and Figure 3 show the raw RSSI values of AP1 obtained at the reference point (0,1) in the corridor of the IT1 building and at the reference point (0,1) in the lobby of the IT2 building, respectively. It can be seen from Figure 2 and Figure 3 that the RSSI value obtained on antenna c is abnormal and cannot be used for positioning, so it needs to be filtered out. At the same time, we perform Gaussian filtering on the RSSI values obtained at antenna a and antenna b. Figure 4 and Figure 5 show the Gaussian filtered RSSI value of AP1 on the effective antenna obtained at the reference point (0,1) in the corridor of the IT1 building and at the reference point (0,1) in the lobby of the IT2 building, respectively. To filter out the residuals of the Gaussian filtering, we further perform the Kalman filtering on the obtained smooth values. Figure 4 and Figure 5 also show the Gaussian-Kalman filtered RSSI values obtained in the two environments.

The algorithm can effectively eliminate the mutation data and noise fluctuation in RSSI fingerprint data. We use the aforementioned data processing method to uniformly process the RSSI values obtained in other indoor environments. The algorithm realizes the accurate and smooth output of RSSI values and is used to establish an accurate fingerprint database, which makes the subsequent positioning more accurate.

#### 3.2.2. Kalman Filtering and Dimension Reduction Processing Based on CSI Amplitude Value

The Wi-Fi signal using the 802.11n transmission protocol can use the OFDM modulation method to modulate the signal to each sub-channel for transmission and to extract CSI on each channel. The CSI data of physical layer is mainly used for wireless network optimization and it is generally difficult to access these data. In recent years, Inter and Atheros network card suppliers have processed some of their network card firmware programs and some relevant organizations have opened corresponding software development packages. We can modify the open-source driver of network card in Linux and WIN system and use debugging mode to obtain CSI data of some wireless network cards. The Intel5300 network card provides sub-carrier level channel measurement for OFDM system, convert the measured value into more abundant multi-path information, and provide more stable measurement value and higher positioning accuracy.

As mentioned above, all software and scripts to read and parse channel measurements are performed by the MATLAB. Finally, a channel matrix with 30 sub-carrier groups is obtained, that is, the 802.11n channel state information is acquired. By giving full play to the advantages of the frequency diversity and space diversity of the MIMO system, while using the characteristics of multiple receiving antennas, the location fingerprint can better reflect the uniqueness of the location. We aggregate CSI in units of receiving antennas under each sub-carrier:(8)$$\begin{array}{cc} H_{csi-1} & =\sum_{m=1}^{p}h_{m1}\\ H_{csi-2} & =\sum_{m=1}^{p}h_{m2}\\ \cdots \\ H_{csi-q} & =\sum_{m=1}^{p}h_{mq} \end{array},$$
where *p* is the number of antennas at the transmitter; $h_{m1},h_{m2},\cdots ,h_{mq}$ are sub-carriers received by receiving antennas 1,2,⋯q, respectively. Figure 6 and Figure 7 show the acquired unprocessed CSI on different receiving antennas in the two environments. It can be seen from Figure 6 and Figure 7 that the CSI value obtained on antenna 3 cannot be used. Here, q=2. Then, the amplitudes are calculated as
(9)$$\;H_{am-i}=\left|H_{csi-i}\right|,\;i=1,2.$$

For each sub-carrier, the mean value of the two amplitudes obtained by the receiving antenna is calculated, so that the dimension of the amplitude becomes 30 dimensions. Figure 8 and Figure 9 show the amplitude of the sub-carriers for 160 measurements in two different indoor environments. The Kalman filtering was performed on the amplitudes of the 30 CSI sub-carriers obtained from different antennas. The obtained CSI value of the Kalman filter is represented by $\hat{H_{am}}$. The ordinate is the magnitude of the amplitude and the abscissa is the time scale. The Kalman filter smooths the waveform further and the CSI amplitude value positioning using the filtering results is feasible. The amplitude of the two sub-carriers at both ends of the obtained sub-carrier are subtracted and dimension-reduced, the effective amplitude fingerprint from an antenna is reduced to 15 dimensions: (10)$$CSI_{AM_{n}}=\frac{1}{2}\left[\hat{H_{am_{30}}}+\hat{H_{am_{1}}},\hat{H_{am_{29}}}+\hat{H_{am_{2}}},\cdots ,\hat{H_{am_{k/2+1}}}+\hat{H_{am_{k/2}}}\right],$$
where k=30,n=1,2,⋯,15. Figure 10 and Figure 11 show the amplitude of the effective CSI sub-carriers on antenna a at reference point (0,1) of AP1 in two different indoor environments.

#### 3.2.3. Linear Transformation and Dimension Reduction of CSI Phase Values

Figure 12 and Figure 13 show the phase of each sub-carrier of the acquired CSI. If the center frequencies of the receiver and transmitter cannot be accurately synchronized (frequency offset), the carrier frequency offset (CFO) will occur. In addition, ADC also produces sampling frequency offset (SFO) [30]. The i-th sub-carrier phase can be expressed as:(11)$${\hat{\varphi }}_{i}=\varphi_{i}-2\pi \frac{k_{i}}{N}\delta +\beta +w_{Z}.$$

Among them, $\varphi_{i}$ is real phase information, δ is timing offset of receiver, β is unknown phase offset, and $w_{Z}$ is phase measurement noise. k denotes the index of the i-th sub-carrier. N is the FFT size.

The existence of the error offset causes the original phase information to not be directly used. Assuming that the value of measurement noise $w_{Z}$ is small, to eliminate the error of δ and β, linear transformation can be used as follows:(12)$$a=\frac{{\hat{\varphi }}_{i}-\bar{\varphi_{1}}}{k_{n}-k_{1}}=\frac{\varphi_{i}-\varphi_{1}}{k_{n}-k_{1}}-\frac{2\pi }{N}\delta ,$$
(13)$$b=\frac{1}{n}\sum_{1\leq j\leq n}{\hat{\varphi }}_{j}=\frac{1}{n}\sum_{1\leq j\leq n}\varphi_{j}-\frac{2\pi \delta }{nN}\sum_{1\leq j\leq n}k_{j}+\beta ,$$
where *a* is the slope of the received response phase. *b* is the offset. If the sub-carrier frequency is symmetric, then $\sum_{j=1}^{n}k_{j}=0$, $\varphi_{i}-ak_{i}-b$ can eliminate the error introduced by δ and β. Because the real phase can not be obtained, only the relationship between the calibrated phase ${\tilde{\varphi }}_{i}$ and the real phase can be obtained, and the difference is a constant multiple Ci related to frequency.
(14)$$\sigma_{{\tilde{\varphi }}_{i}}^{2}=c_{i}\sigma_{\varphi_{i}}^{2},$$
(15)$$\;c_{i}=1+2\frac{k_{i}^{2}}{{\left(k_{n}-k_{1}\right)}^{2}}+\frac{1}{n}.$$

Figure 14 and Figure 15 give the phase information processed linearly, which is much more stable than the original phase in Figure 12 and Figure 13. The phase received by the receiving end has random errors due to time delay and multi-path effects, which are the main factors affecting the accuracy of phase estimation, and they need to be eliminated to improve the accuracy. The CSI is processed by a linear transformation, which effectively eliminates the phase error and obtains a higher-precision estimated phase. It is possible to establish more accurate indoor fingerprints, and, for each sub-carrier, the three calibration phases are subtracted to make the fingerprint of phase information 30 dimensions: (16)$$H_{ph}=H_{ph-1}-H_{ph-2},$$
where $H_{ph-i}={\tilde{\varphi }}_{i},i=1,2$.

After analyzing the trend of CSI phase, this paper proposes a data dimension reduction method.This method is easy to operate and reduces the calculation at the same time. Since the information contained in the adjacent sub-carriers is nearly similar, we remove the amplitude of the even-numbered adjacent sub-carriers according to this feature, and only preserve the phase information of the odd-numbered sub-carriers. Finally, the effective CSI obtained by dimension reduction can be used to establish offline CSI fingerprint database. Dimensionality reduction processing is carried out for the phase information of the two sub-carriers symmetric about the center sub-carrier, and the phase information from an antenna is reduced to 15 dimension:(17)$$CSI_{PH_{n}}=\left[\hat{H_{ph_{1}}},\hat{H_{ph_{3}}},\cdots ,\hat{H_{ph_{k/2}}},\cdots ,\hat{H_{ph_{k-3}}},\hat{H_{ph_{k-1}}}\right],\;k=30,\;n=1,2,\cdots ,15.$$

Figure 16 and Figure 17 show the variation trend of CSI phases in different dimensions under different indoor environments. It can be seen that, even though the phase dimension of the sub-carrier is 15, the phase change trend can be clearly presented.

#### 3.2.4. Location Fingerprint Generation Based on Data Fusion

By the aforementioned processing of RSSI, CSI amplitude, and CSI phase information, we obtain the fingerprints containing 2 × 1 dimensional RSSIs, 2 × 15 dimensional CSI amplitude, and 2 × 15 dimensional CSI phase information, respectively. $\left(x_{m},y_{m}\right)$ can give the uniqueness of the position coordinates.
(18)$$\left(x_{m},y_{m}\right)=\left[RSSI_{i}\;CSI_{i}AM_{n}\;CSI_{i}PH_{n}\right],$$
where *m* represents the number of packets collected. *i* represents the number of antennas and i=1,2. n=1,2,⋯,15. In addition, when there are multiple access points (APs) in the positioning system, they can be aggregated for the participate of location fingerprints.
(19)$$\left(x_{RP},y_{RP}\right)=\left[\begin{array}{c} {RSSI_{i}}_{1},{CSI_{i}AM_{n}}_{1},{CSI_{i}PH_{n}}_{1}\\ {RSSI_{i}}_{2},{CSI_{i}AM_{n}}_{2},{CSI_{i}PH_{n}}_{2}\\ \cdots \\ {RSSI_{i}}_{s},{CSI_{i}AM_{n}}_{s},{CSI_{i}PH_{n}}_{s} \end{array}\right],$$
where *s* represents the number of APs, and, in this paper, s=3. $\left(x_{RP},y_{RP}\right)$ is the position coordinates of the reference point stored in the fingerprint database during the offline phase. Online phase, the user’s fingerprint information is compared with the fingerprint database to obtain several different areas. In this paper, we select *t* test points for fingerprint matching, and the coordinate value of the test point is $\left(x_{Test},y_{Test}\right)$. The WKNN algorithm is used to aggregate the above regions and to select the optimal estimation location.
(20)$$\left(x_{Test},y_{Test}\right)=\left[\begin{array}{c} {RSSI_{i}}_{1},{CSI_{i}AM_{n}}_{1},{CSI_{i}PH_{n}}_{1}\\ {RSSI_{i}}_{2},{CSI_{i}AM_{n}}_{2},{CSI_{i}PH_{n}}_{2}\\ \cdots \\ {RSSI_{i}}_{t},{CSI_{i}AM_{n}}_{t},{CSI_{i}PH_{n}}_{t} \end{array}\right].$$

## 4. Experimental Environment and Performance Evaluation

### 4.1. Experimental Environment

Figure 18 and Figure 19 verify the performance of the proposed algorithm in the corridor on the 3rd floor of IT1 building and the hall on the 1st floor of IT2 building in KNU. The floor plan is shown in Figure 20 and Figure 21.

The receiver of this paper is a laptop with ubuntu14.04.4 operating system and intel5300 wireless network card. The network card is attached with three dual-frequency external antennas of 2.4 G/5 GHz. The green dots in the figure indicate the positions of reference points. The accuracy of RSSI signal and CSI signal is the highest at a height of 1 m from the ground, so the distance between the AP antenna and the ground is 1 m. When collecting RSSI signal and CSI signal data, the tester holds a portable notebook device and stands at the reference point for data collection. The reference point receives signals from three APs. Each reference point collects RSSI data and CSI data for 30 min and saves them to the PC. The RSSI data and CSI data are extracted by MATLAB, and the original RSSI data and CSI data are filtered at the same time. Take the effective RSSI value and the effective CSI value to establish the database as the feature vector of the point. The experimental starts from the lower-left corner, and its coordinates are (0,0). The grid size of the fingerprint database is 1 m. In the corridor environment, fingerprint information of 24 locations are collected, and 27 locations are collected in the lobby environment. The performance of the algorithm is evaluated by the distance error between the estimated position of the reference point and the real position.

### 4.2. Performance Evaluation

In the indoor environment, there are many factors that affect the positioning results, such as the number of people in the positioning area and their status, static or moving, receiver location and antenna orientation, and the influence of other indoor equipment movements, etc. We consider these conditions that are difficult to control and evaluate the generality of the proposed algorithm. To weaken these external factors, the same set of location data is processed to compare the performance of the location algorithm.

#### 4.2.1. Impact of the Number of Packets

During the construction of fingerprint database, the number of samples makes a great impact on the positioning performance. For the similarity comparison method of fingerprint database, the location accuracy largely depends on whether the fingerprint samples in the fingerprint database can accurately describe the distribution of the overall fingerprint or not. For this experiment, each unit area is sampled for 5 min, and nearly 1000–3000 CSI samples can be sampled for each AP. In the case of the same environment parameters, we selected different numbers of sample sets for training to verify the existing indoor location method. We also investigated the influence of the different sample set number in the proposed fingerprint location method on the positioning performance. The sampling frequency of sampled data is equal to 1500 packets/s.

Figure 22 and Figure 23 show the impact on positioning performance when the number of samples in the process of establishing a fingerprint database is 100, 500, 1000, and 2000, respectively. In the two experimental scenarios, we select three APs to establish fingerprints and set k=3 in the WKNN algorithm. The number of samples in the on-line phase is about 50, 250, 500, and 1000, respectively. When the sampling rate is increased from 100 to 500 packets, as shown in the figure. When it increased from 500 to 1000, the positioning accuracy showed a trend of improvement. When it is between 1000 and 2000, the positioning accuracy has not improved. Based on the above experimental results, the positioning accuracy of 1000 packets achieved the highest score under certain conditions. We also compared the average distance errors at reference point (1.1) and reference point (4,2), as shown in Figure 22 and Figure 23. It can be seen from Figure 22 and Figure 23 that selecting 1000 data packets at different reference points to build a database obtained more accurate positioning.

#### 4.2.2. Comparison with Existing Fingerprint Location Methods

In the positioning error analysis, cumulative distribution function (CDF), standard deviation, and average error were used to analyze the performance of the positioning method. In two indoor environments, the proposed method is compared with the RSSI-based fingerprint positioning system and two CSI-based fingerprint positioning systems (FIFS and CSI-MIMO). In the test process, WKNN is used as fingerprint matching algorithm and three APs are used to generate fingerprint. We use Euclidean distance to estimate the similarity measure between the reference point (RP) coordinates $\left(X_{RP},Y_{RP}\right)$ and the test point (TP) coordinates $\left(X_{TP},Y_{TP}\right)$. The formula is as follows:(21)$$\hat{d}=\sqrt{{\left(X_{RP}-X_{TP}\right)}^{2}-{\left(Y_{RP}-Y_{TP}\right)}^{2}}.$$

The CDFs of location distance error in the two scenarios are shown in Figure 24 and Figure 25. Figure 24 is the CDF figure obtained from the experiment in the corridor of IT-1 building. It can be seen from the figure that RSSI-based approach is vulnerable to environmental interference and is very unstable, and the error is the largest among the four positioning methods. FIFS and CSI-MIMO use channel state information as fingerprint eigenvalues. Due to the finer granularity of channel state information, the multi-path interference can be suppressed to some extent and the positioning accuracy of these two methods is higher than that of the RSSI-based approach. The proposed location method, based on FIFS and CSI-MIMO, and according to the characteristics of multiple antennas, improves the positioning accuracy by improving the fingerprint feature dimension. The red curve in Figure 24 shows that the probability of positioning error within 1.5 m reaches 60%, which is 33.33% higher than that of CSI-MIMO and 51.24% higher than that of FIFS. Figure 25 is the CDF diagram of the experiment in the lobby of IT-2 building. Compared with Figure 24, the accuracy rate is improved, which indicates that the environment affected the accuracy of the experiment. However, it can be seen from Figure 25 that the positioning accuracy of the proposed method is still higher than those of the FIFS and CSI-MIMO methods. The probability of positioning error within 1.5 m is 81.4%, which is 4.7% higher than that of CSI-MIMO and 41.4% higher than that of FIFS. Finally, the proposed method, FIFS method, and CSI-MIMO method are compared in two experimental environments by using the average positioning error, standard deviation. The results are shown in Table 3 and Table 4.

It can be seen from the table, in terms of the average positioning error, the performance of the proposed method is about 1.171 m in the corridor environment, which is about 12.6% higher than CSI-MIMO and 36.9% higher than that of FIFS. The average positioning error of the RSSI-based method is about 2.122 m, it is lower than the positioning accuracy of the proposed method; In the lobby experimental scenario, the proposed method is about 1.094 m, which is 15.9% and 61.5% higher than CSI-MIMO and FIFS, respectively. The average positioning error of the RSSI-based method is about 2.078 m. Through the series of experiments, we confirmed that the method proposed in this paper was effective in improving the accuracy of positioning. The positioning method proposed in this paper can effectively improve the positioning accuracy by the following reasons. The reasons for the proposed algorithm to improve accuracy are as follows. Firstly, the fingerprints in the fingerprint database are integrated with RSS and CSI, which are not based on CSI or RSS alone; secondly, the fingerprint features are preprocessed effectively in the offline phase, and the integrity of the data is preserved; thirdly, WKNN is used to reduce the positioning error. In addition, since the proposed method performs the dimensional reduction for the fingerprint, its computational complexity is also improved over the CSI-MIMO and FIFS.

## 5. Conclusions and Future Work

In this paper, a hybrid fingerprint algorithm is proposed to improve the positioning accuracy of RSSI-based and CSI-based fingerprint location methods. In the offline phase, the amplitude values of RSSI and CSI are eliminated and filtered, and the phase value of CSI is introduced by linear transformation. This paper also proposes a method to reduce the fingerprint feature dimension of high-dimensional data CSI obtained in OFDM-MIMO system. Finally, combined with the effective RSSI, effective CSI amplitude value, and CSI phase value, a novel fingerprint database containing more abundant indoor location information was established. In the online phase, the fingerprint data is matched according to the WKNN algorithm, which effectively improves the positioning accuracy. This paper uses MATLAB to verify the effectiveness and superiority of the algorithm in two indoor environments. Compared with existing algorithms, the algorithm based on RSS and CSI hybrid fingerprints can improve positioning accuracy. At the same time, because additional hardware is not necessary for the proposed algorithm to perform positioning, its applicability is also high. Future research directions include the evaluation of the performance of the proposed algorithm in a three-dimensional indoor environment, an indoor laboratory environment with more obstacles and personnel flows.

## Figures and Tables

**Figure 1 sensors-21-02769-f001:**
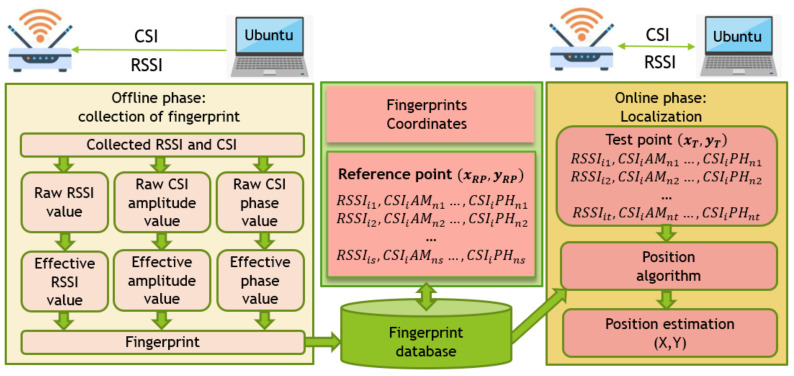
Indoor fingerprint localization architecture.

**Figure 2 sensors-21-02769-f002:**
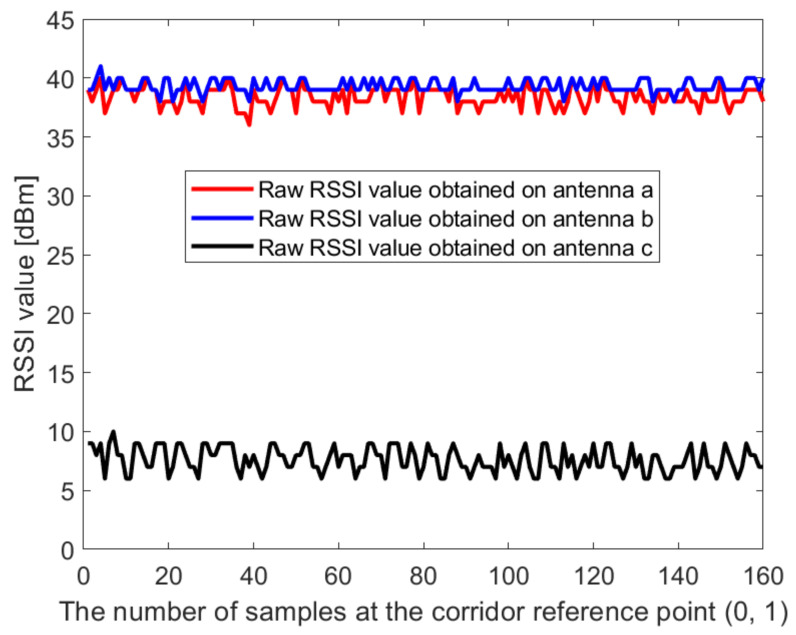
The raw RSSI value of access point AP1 obtained at the reference point (0,1) in the corridor of IT1 building.

**Figure 3 sensors-21-02769-f003:**
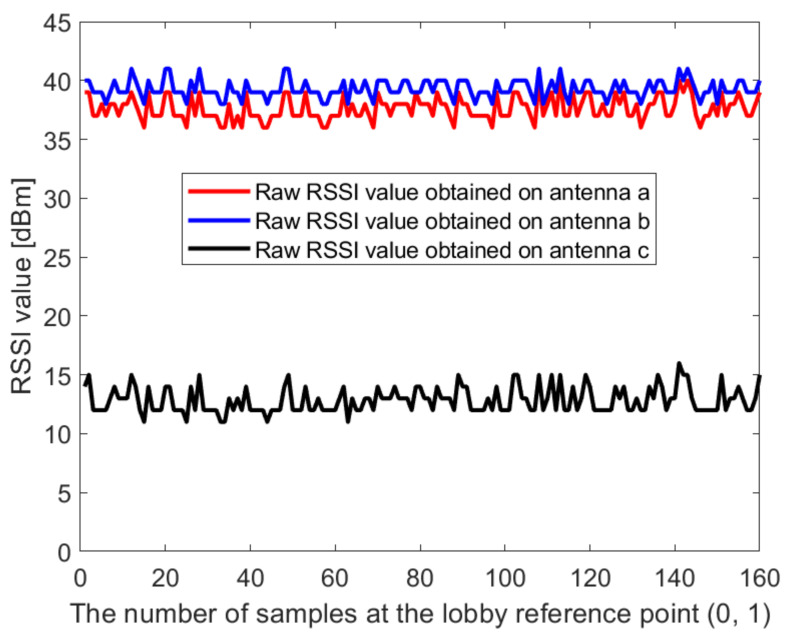
The raw RSSI value of AP1 obtained at the reference point (0,1) in the lobby of IT2 building.

**Figure 4 sensors-21-02769-f004:**
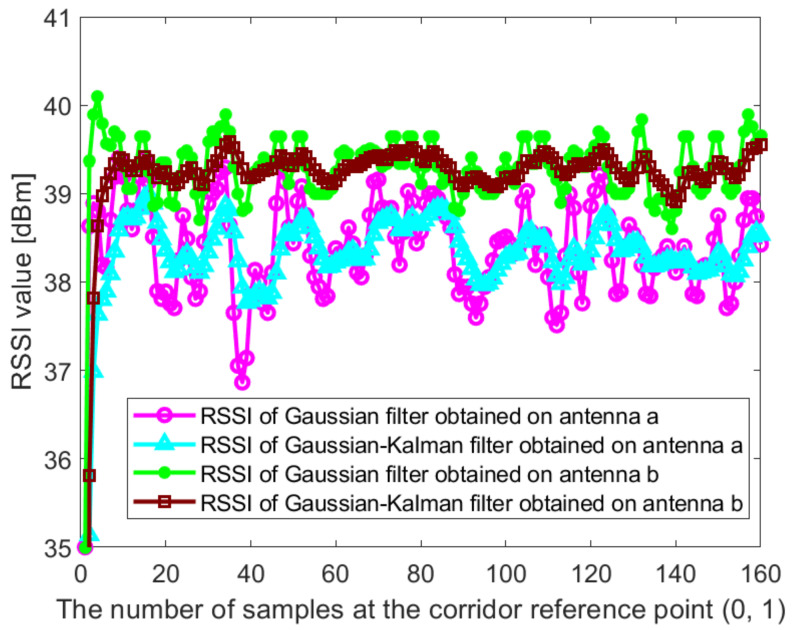
The Gaussian filtered RSSI value and the Gaussian-Kalman filtered RSSI value of AP1 on the effective antenna obtained at the reference point (0,1) in the corridor of IT1 building.

**Figure 5 sensors-21-02769-f005:**
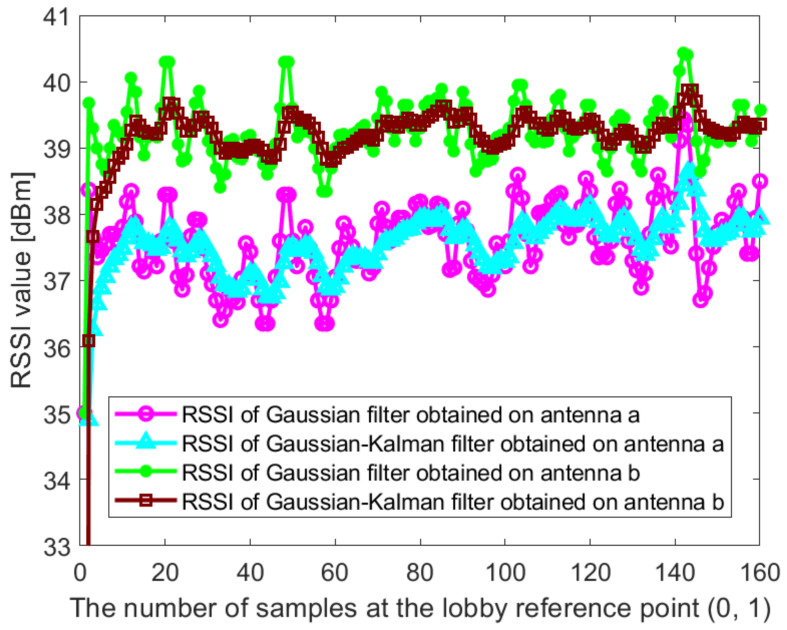
The Gaussian filtered RSSI value and the Gaussian-Kalman filtered RSSI value of AP1 on the effective antenna obtained at the reference point (0,1) in the lobby of IT2 building.

**Figure 6 sensors-21-02769-f006:**
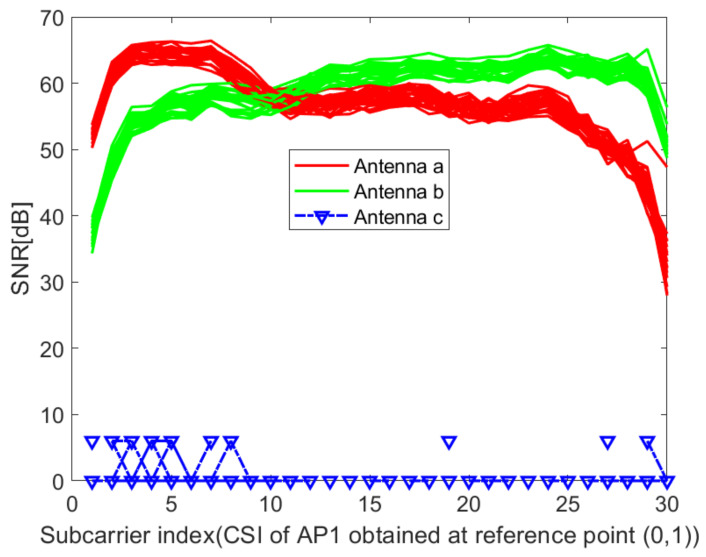
Raw CSI obtained on different receiving antennas at reference point (0,1) of AP1 in the corridor of IT1 building.

**Figure 7 sensors-21-02769-f007:**
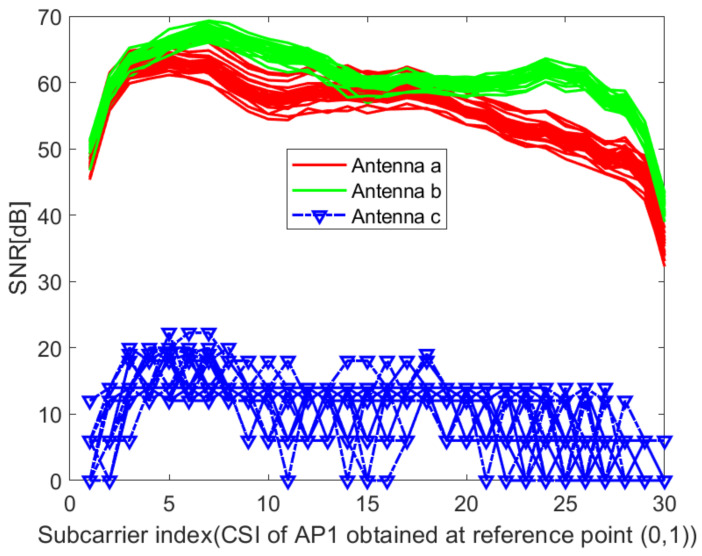
Raw CSI obtained on different receiving antennas at reference point (0,1) of AP1 in the lobby of IT2 building.

**Figure 8 sensors-21-02769-f008:**
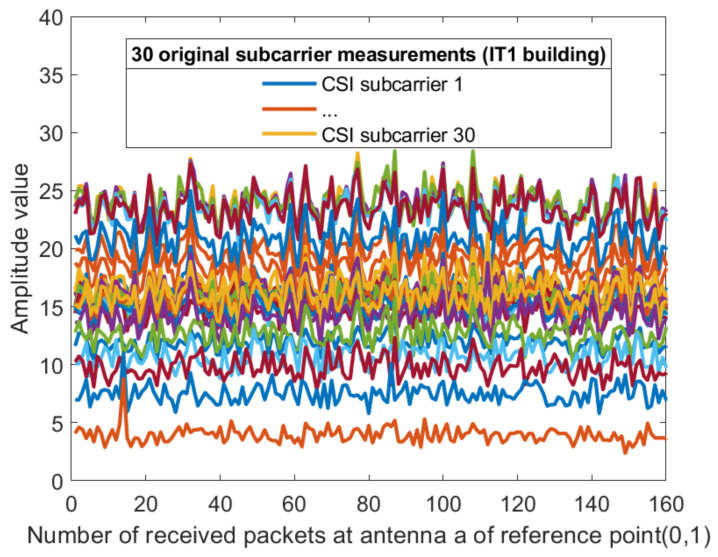
The amplitude of the 30 CSI sub-carriers on antenna a at reference point (0,1) of AP1 in the corridor of IT1 building.

**Figure 9 sensors-21-02769-f009:**
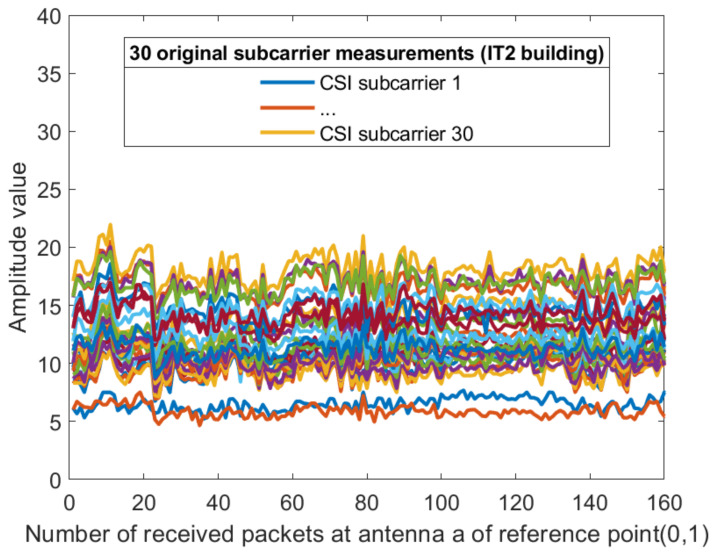
The amplitude of the 30 CSI sub-carriers on antenna a at reference point (0,1) of AP1 in the lobby of IT2 building.

**Figure 10 sensors-21-02769-f010:**
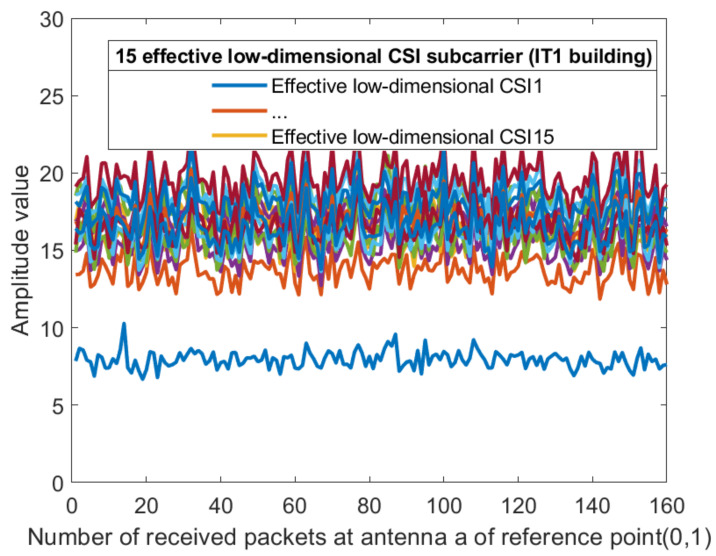
The amplitude of the effective CSI sub-carriers on antenna a at reference point (0,1) of AP1 in the corridor of IT1 building.

**Figure 11 sensors-21-02769-f011:**
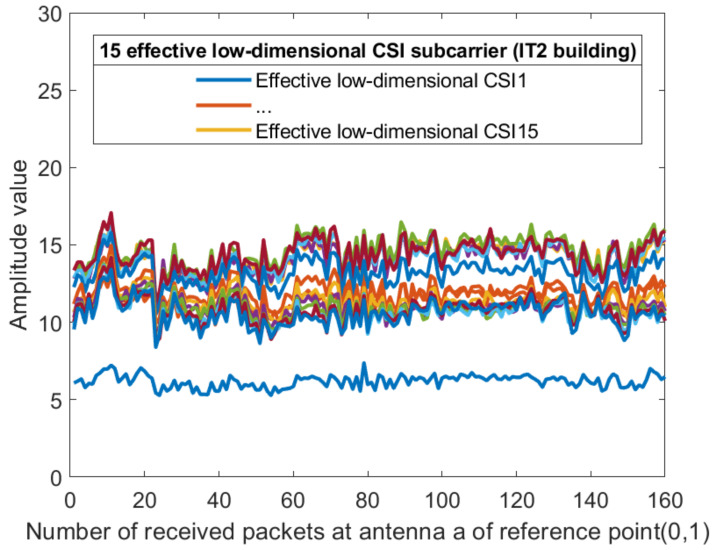
The amplitude of the effective CSI sub-carriers on antenna a at reference point (0,1) of AP1 in the lobby of IT2 building.

**Figure 12 sensors-21-02769-f012:**
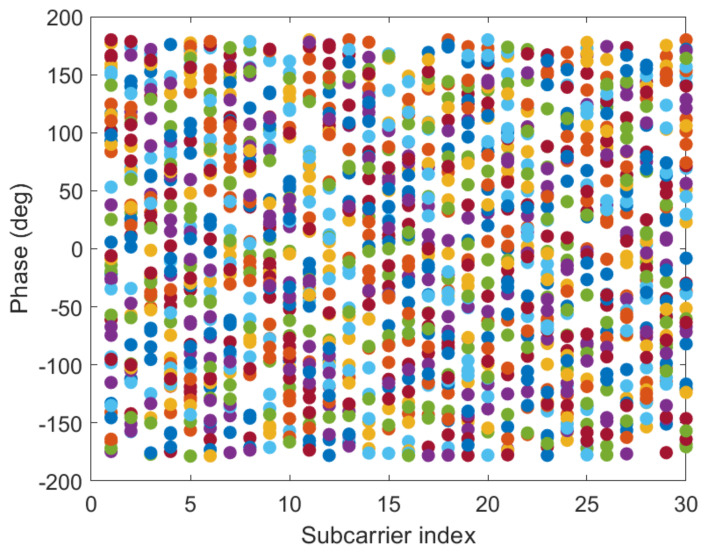
The phases of the 30 CSI sub-carriers on antenna a at reference point (0,1) of AP1 in the corridor of IT1 building.

**Figure 13 sensors-21-02769-f013:**
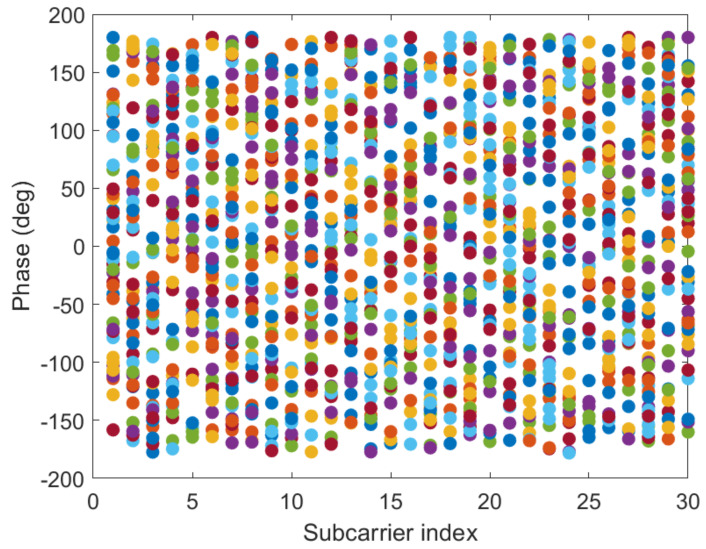
The phases of the 30 CSI sub-carriers on antenna a at reference point (0,1) of AP1 in the lobby of IT2 building.

**Figure 14 sensors-21-02769-f014:**
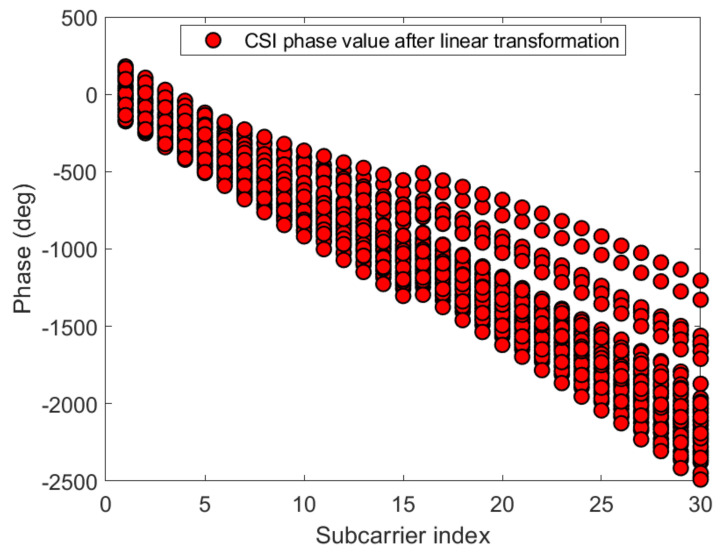
The phases after linear transformation of the 30 CSI sub-carriers on antenna a at reference point (0,1) of AP1 in the corridor of IT1 building.

**Figure 15 sensors-21-02769-f015:**
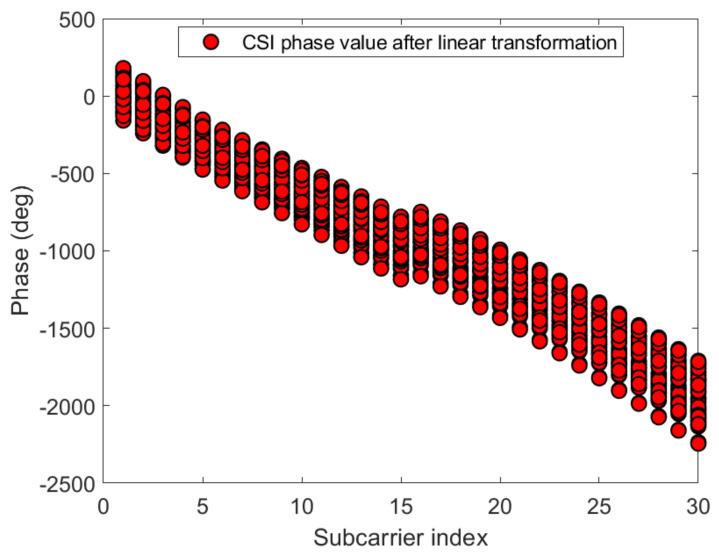
The phases after linear transformation of the 30 CSI sub-carriers on antenna a at reference point (0,1) of AP1 in the lobby of IT2 building.

**Figure 16 sensors-21-02769-f016:**
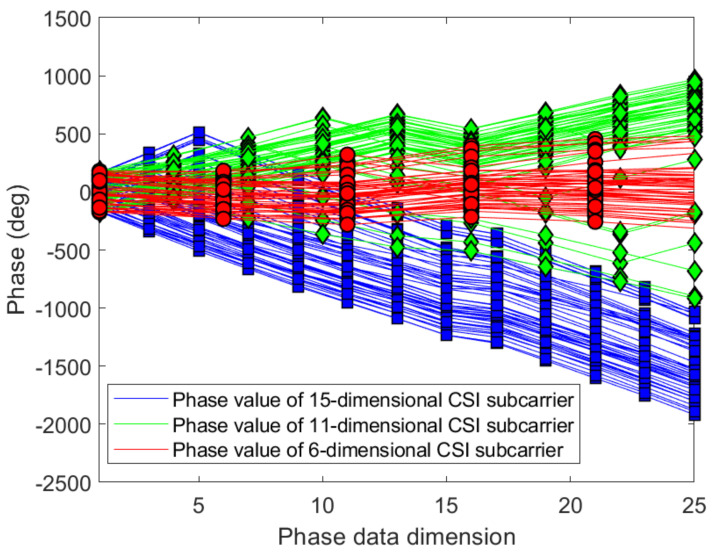
Compare the phases change trends with different dimensions on antenna a at reference point (0,1) of AP1 in the corridor of IT1 building.

**Figure 17 sensors-21-02769-f017:**
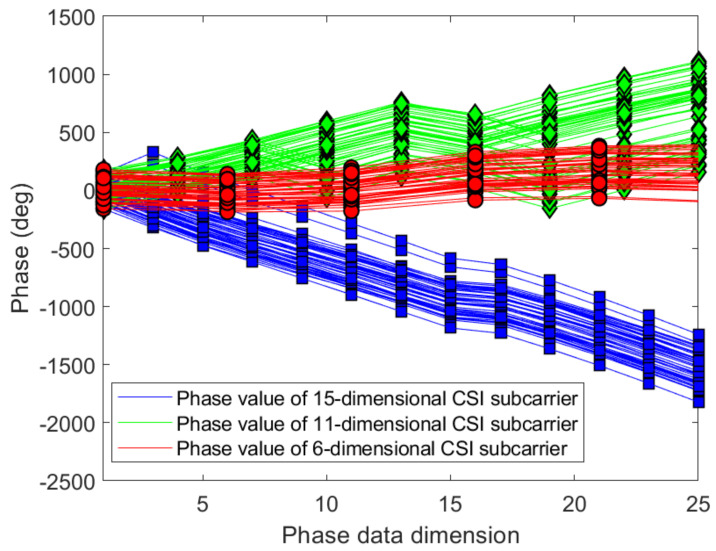
Compare the phases change trends with different dimensions on antenna a at reference point (0,1) of AP1 in the lobby of IT2 building.

**Figure 18 sensors-21-02769-f018:**
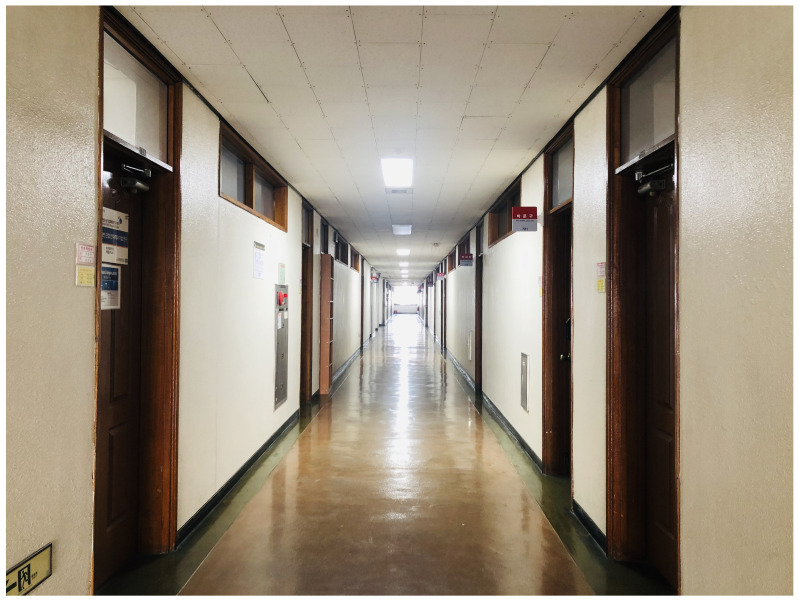
Real corridor environment on the 3rd floor of Kyungpook National University (KNU) IT-1 building.

**Figure 19 sensors-21-02769-f019:**
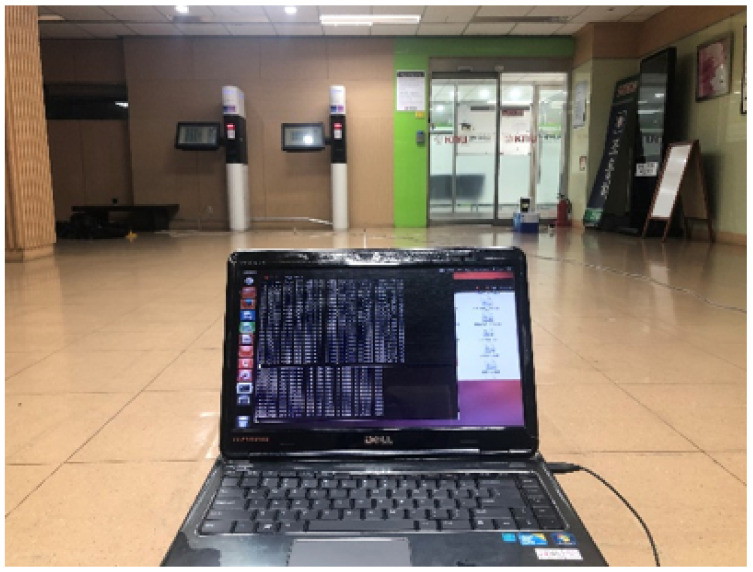
Real lobby environment on the 1st floor of KNU IT-2 building.

**Figure 20 sensors-21-02769-f020:**
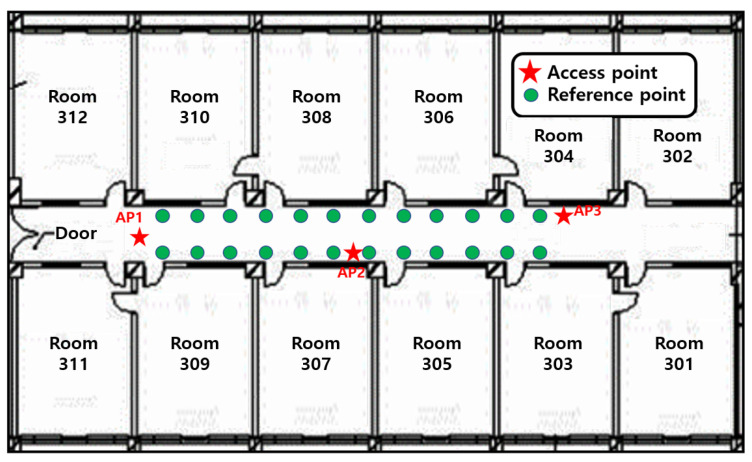
Floor plan of the corridor on the 3rd floor of KNU IT-1 building.

**Figure 21 sensors-21-02769-f021:**
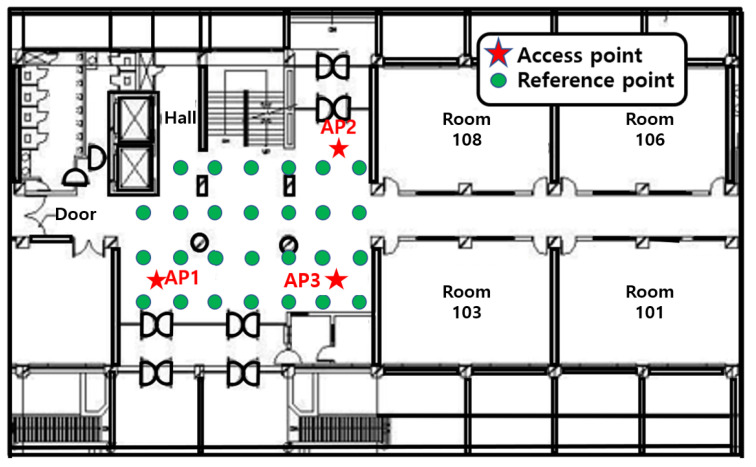
Floor plan of the lobby on the 1st floor of KNU IT-2 building.

**Figure 22 sensors-21-02769-f022:**
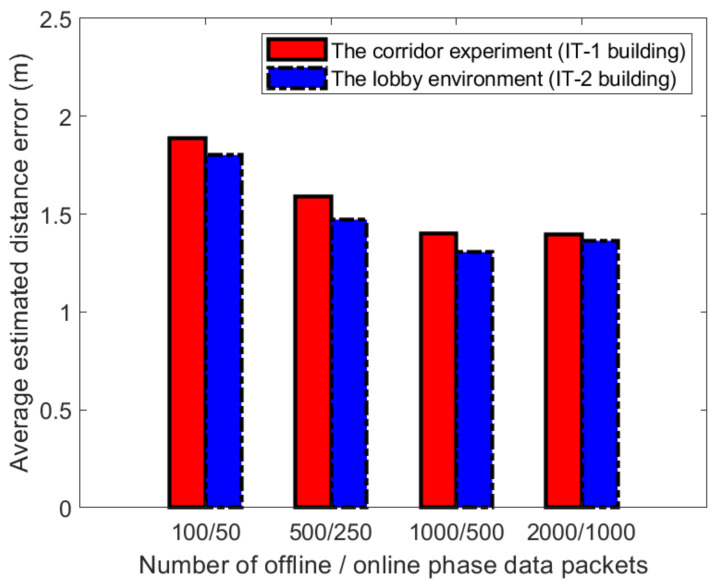
Distance estimation errors at reference points (1,1) in two different experimental environments.

**Figure 23 sensors-21-02769-f023:**
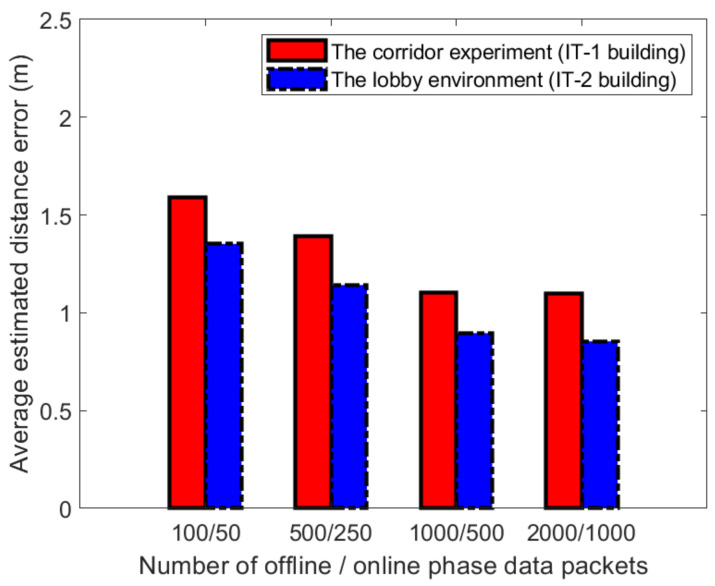
Distance estimation errors at reference points (4,2) in two different experimental environments.

**Figure 24 sensors-21-02769-f024:**
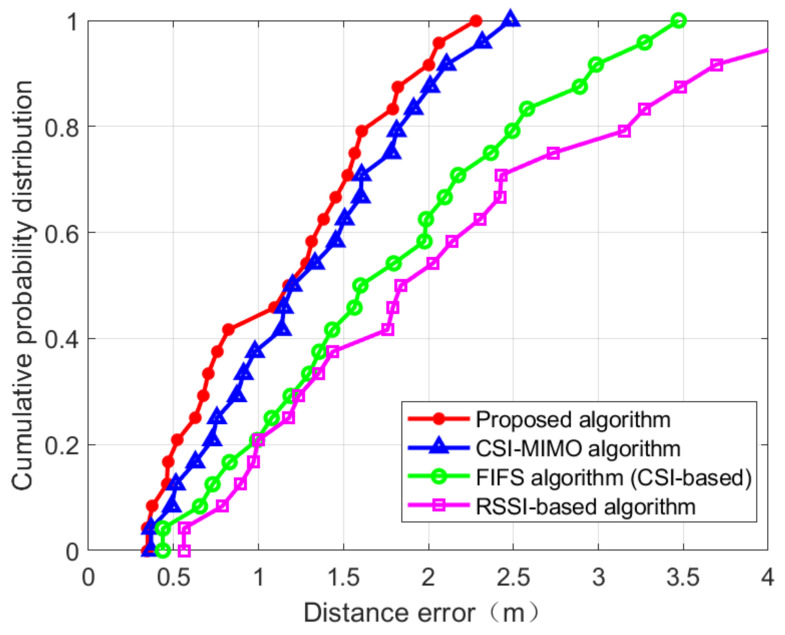
Comparing the cumulative distribution function (CDF) value of localization error of four algorithm in IT-1 building.

**Figure 25 sensors-21-02769-f025:**
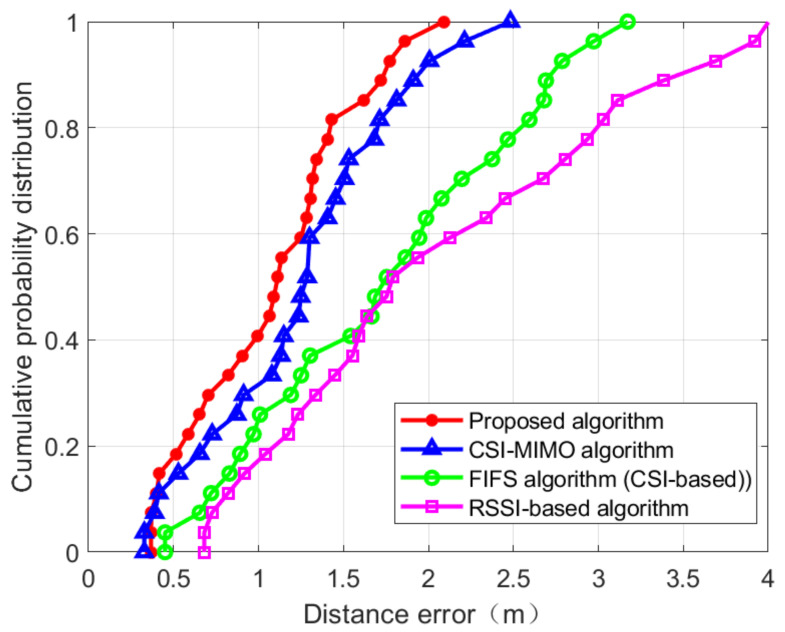
Comparing the CDF value of localization error of four algorithm in IT-2 building.

**Table 1 sensors-21-02769-t001:** Compare the difference between received signal strength indication (RSSI) and channel state information (CSI).

Category	RSSI	CSI
Layer	MAC layer	Physical layer
Granularity	Coarse-grained	Fine-grained
Time resolution	Packet	Multipath signal cluster
Frequency resolution	None	Subcarrier
Stability	Low	High
Dimension	One dimension	High dimension
Power consumption	Low	High
Mathematical value	Real number	Complex number
Universality	All Wi-Fi devices	Some Wi-Fi devices

**Table 2 sensors-21-02769-t002:** Channel state information (CSI).

Data Information	Properties
Bfee-count	Number of Bfee count beamforming
Nrx	Number of receiver antennas
Ntx	Number of transmitter antennas
rssi-a,rssi-b,rssi-c	RSS of each receiving antenna
rate	Transmission rate of each data packet
noise	noise
CSI	CSI is a 3-dimensions array of Nrx × Ntx ×30

**Table 3 sensors-21-02769-t003:** Comparison of four indoor fingerprint location algorithms (corridor of IT-1 building).

Fingerprint Algorithm	Average Distance Error (m)	Standard Deviation (m)
RSSI-based algorithm	2.122 m	1.097 m
CSI-based algorithm (FIFS)	1.802 m	0.853 m
CSI-MIMO algorithm	1.319 m	0.605 m
Proposed algorithm	1.171 m	0.587 m

**Table 4 sensors-21-02769-t004:** Comparison of indoor fingerprint location algorithms (lobby of IT-2 building).

Fingerprint Algorithm	Average Distance Error (m)	Standard Deviation (m)
RSSI-based algorithm	2.078 m	1.007 m
CSI-based algorithm (FIFS)	1.767 m	0.781 m
CSI-MIMO algorithm	1.269 m	0.559 m
Proposed algorithm	1.094 m	0.488 m

## Data Availability

Not applicable.

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
