# Peer review of "An Enhanced Indoor Positioning Algorithm Based on Fingerprint Using Fine-Grained CSI and RSSI Measurements of IEEE 802.11n WLAN"

_sensors, 2021, doi:10.3390/s21082769_

Round 1

Reviewer 1 Report

In this paper, a hybrid fingerprint algorithm is proposed to improve the positioning accuracy of RSSI-based and CSI-based fingerprint location methods. I feel that this paper is interesting and enjoys a great amount of experiments and measurements. Thus, I recommend acceptance after minor revision.  The minor issues were listed as below.

First, I wonder the normalization mechanism is required because the scale is different between the CSI phase and RSS power. It would be better to address this issue and provide some comparison.  It would be better to mention the difference with the related works. Finally, the English should be checked carefully. There are some typos. For example, page 3 line 103, approach Including should be approach including.   

Reviewer 2 Report

This paper is interesting as it proposed a method that combined the fingerprint of RSS and CSI to overcome the problem of WiFi signal fluctuation and multipath. However, it is inadequate in the elaboration and experimental verification of the key issues.

  • It is well known that the CSI information achieves higher robustness than traditional RSS fingerprinting. So what are the benefits of combining the two?
  • In Fig. 4 and 5, the Gaussian filtered RSSI and Kalman filtered RSSI values are depicted. What is the purpose of this treatment? What is the value in practical applications? In practical situations, the variation of WiFi signal is difficult to predict due to the complexity of indoor environments.
  • The paper should also be supplemented with the actual positioning errors, not just the signal ranging errors.
  • In page 1, line 27, “ … GPS signals … vulnerable to multipath effects, “ is this the main reason that GPS signals do not work indoors?
  • In page 3, line 96, there is an extra comma in the sentence.
  • It is also suggested to supply the range accuracy of CSI signals in different indoor situations, including the case of occlusion by indoor objects.

Reviewer 3 Report

The authors propose a hybrid fingerprint indoor location method based on combining RSS and CSI.  Kalman filter and a Gaussian function are used for signal preprocessing in the  offline phase. The information is used for establishing of a hybrid fingerprint database with reduced dimensionality. The low complexity WKNN algorithm is applied during the online positioning stage. Experimental results show that the proposed algorithm shows higher positioning accuracy and smaller positioning error compared with conventional methods.

The manuscript is mostly well written. However, the presentation may be improved.

The authors are suggested to consider introducing of the expermental environment (subtitle 4.1) before figures Fig 2. and Fig 3. It would be easier for the reader to understand the meaning of the contents of figures. 

The meaning of different lines in figures Fig. 7, Fig. 9, Fig. 10 and Fig 11 should be more explicitly defined. Also showing a smaller number of selected examples of subcarriers may be considered to aviod ambiguity. 

Round 2

Reviewer 2 Report

The authors have addressed the comments satisfactorily. This is an interesting and solid work for indoor positioning. Well done.